# Aerobic Exercise and Neuropathic Pain: Insights from Animal Models and Implications for Human Therapy

**DOI:** 10.3390/biomedicines11123174

**Published:** 2023-11-29

**Authors:** Jorge Ruimonte-Crespo, Gustavo Plaza-Manzano, María José Díaz-Arribas, Marcos José Navarro-Santana, José Javier López-Marcos, Raúl Fabero-Garrido, Tamara Seijas-Fernández, Juan Antonio Valera-Calero

**Affiliations:** 1Department of Radiology, Rehabilitation and Physiotherapy, Complutense University of Madrid, 28040 Madrid, Spain; jruimont@ucm.es (J.R.-C.); mjdiazar@med.ucm.es (M.J.D.-A.); marconav@ucm.es (M.J.N.-S.); josejalo@ucm.es (J.J.L.-M.); rfabero@ucm.es (R.F.-G.); tamaraseijasfernandez@gmail.com (T.S.-F.); juavaler@ucm.es (J.A.V.-C.); 2Grupo InPhysio, Instituto de Investigación Sanitaria del Hospital Clínico San Carlos (IdISSC), 28040 Madrid, Spain; 3Faculty of Life and Natural Sciences, Nebrija University, 28015 Madrid, Spain

**Keywords:** neuropathic pain, pain, aerobic exercise, therapeutic exercise

## Abstract

This narrative review explores the complex relationship between aerobic exercise (AE) and neuropathic pain (NP), particularly focusing on peripheral neuropathies of mechanical origin. Pain, a multifaceted phenomenon, significantly impacts functionality and distress. The International Association for the Study of Pain’s definition highlights pain’s biopsychosocial nature, emphasizing the importance of patient articulation. Neuropathic pain, arising from various underlying processes, presents unique challenges in diagnosis and treatment. Our methodology involved a comprehensive literature search in the PubMed and SCOPUS databases, focusing on studies relating AE to NP, specifically in peripheral neuropathies caused by mechanical forces. The search yielded 28 articles and 1 book, primarily animal model studies, providing insights into the efficacy of AE in NP management. Results from animal models demonstrate that AE, particularly in forms like no-incline treadmill and swimming, effectively reduces mechanical allodynia and thermal hypersensitivity associated with NP. AE influences neurophysiological mechanisms underlying NP, modulating neurotrophins, cytokines, and glial cell activity. These findings suggest AE’s potential in attenuating neurophysiological alterations in NP. However, human model studies are scarce, limiting the direct extrapolation of these findings to human neuropathic conditions. The few available studies indicate AE’s potential benefits in peripheral NP, but a lack of specificity in these studies necessitates further research. In conclusion, while animal models show promising results regarding AE’s role in mitigating NP symptoms and influencing underlying neurophysiological mechanisms, more human-centric research is required. This review underscores the need for targeted clinical trials to fully understand and harness AE’s therapeutic potential in human neuropathic pain, especially of mechanical origin.

## 1. Introduction

### 1.1. Definition of Pain

Pain is a predominant challenge for clinicians and researchers across various specialties, given its potential to significantly impair functionality and induce distress. Owing to progressive insights in pain research and neurophysiology, the International Association for the Study of Pain (IASP) revised the definition of pain in 2020. It is now articulated as “an aversive sensory and emotional experience, either akin to or associated with actual or potential tissue damage” [1].

This refined definition underscores the multifaceted nature of pain, emphasizing its emergence from intricate biopsychosocial interactions. It can manifest adaptively or maladaptively. Moreover, it accentuates the imperative of acknowledging a patient’s articulation of pain, irrespective of empirical validation. Broadly, the IASP categorizes pain into three distinct types: nociceptive pain (originating from the activation of nociceptors), neuropathic pain (resulting from anomalies or pathologies within the somatosensory system), and nociplastic pain (pain originating from changed nociception, even in the absence of apparent or imminent tissue harm that would activate peripheral nociceptors, or without any signs of disease or lesions in the somatosensory system that could cause the pain) [1,2].

While this classification is rooted in foundational biological mechanisms, it is important to recognize that it does not, on its own, dictate a specific diagnosis. The classification provides a framework for understanding the underlying pathophysiology of pain, but accurate diagnosis typically requires corroborative functional test results. These tests help in clarifying the diagnosis by revealing the functional impact and specific characteristics of the pain. Additionally, it is noteworthy that these pain categories are not mutually exclusive; many clinical scenarios may exhibit overlapping features, necessitating a comprehensive diagnostic approach that integrates both biological mechanisms and functional test outcomes.

### 1.2. Insight into Neuropathic Pain

Despite its ostensibly straightforward definition, the experience of neuropathic pain (NP) arises from a myriad of underlying processes. This complexity not only complicates its study but can also influence the efficacy of therapeutic interventions.

The duration of neuropathic symptoms and signs can be categorized as either acute (lasting less than 3 months) or chronic (persisting for more than 3 months). Additionally, based on its etiological origin, NP can be delineated into central NP (resulting from afflictions within the central nervous system, CNS) or peripheral NP (stemming from disorders in the peripheral nervous system, PNS) [3].

Examples of peripheral neuropathic pain encompass conditions like trigeminal neuralgia, painful radiculopathy, and diverse painful peripheral polyneuropathies (attributable to metabolic, infectious, toxic or hereditary causes, etc.), as well as those due to direct injuries. Central NP, on the other hand, includes pain associated with spinal cord injuries, brain traumas, strokes, and multiple sclerosis, to name a few [3].

A significant challenge with this NP definition is its insistence on objectively identifying an aberration in the somatosensory system. In numerous instances, despite the evident presence of NP, objective results remain elusive. Consequently, in such scenarios, NP can only be termed as “possible” rather than “probable” or “definite” [4,5]. This predicament underscores the reliance on clinical diagnoses in routine practice, which are often based on the patient’s pain characteristics in the absence of objective tests. Moreover, the term “somatosensory system” lacks precise delineation, encompassing both nociceptive and non-nociceptive afferent pathways, in addition to various pain modulation systems [5].

Within the ambit of peripherally originated NP, a spectrum of symptoms can be identified, with their plausible etiology rooted in neuropathic processes. Notably, these include friction-induced allodynia, dysesthesias, persistent burning sensations, and episodic electric or stabbing pains [4]. The specific subtypes of NP that manifest is of paramount importance, as they can significantly influence treatment outcomes [5]. NP often coexists with other neuropathic characteristic symptoms and signs that do not strictly qualify as NP. These include paresthesias, hypoesthesias, hyperesthesias, muscle weakness, and alterations in deep tendon reflexes. Additionally, NP frequently co-occurs with sleep disturbances, anxiety and depression [4,5].

While these manifestations can be discerned through physical examination and patient history, they guide the diagnostic assessment toward NP but do not conclusively confirm its presence.

### 1.3. Assessment of Neuropathic Pain

There are currently a series of validated questionnaires that allow NP to be detected in a reliable, simple and economical way, although not 100% accurate [5]. Among all of them, the references are the Leeds Assessment of Neuropathic Symptoms and Signs (S-LANSS), the Neuropathic Pain Questionnaire, the four-question Neuropathic Pain Questionnaire and PainDETECT [5]. All of them present a series of questions that the patient must answer. In addition, some of them present a sensory examination, which must be carried out by a health professional. In addition, we have questionnaires for clinical monitoring of NP, among which the Neuropathic Pain Symptom Inventory stands out [5].

Quantitative sensory tests are also among the non-invasive tools for detecting NP. These allow us to find dysfunctions in the somatosensory system by measuring stimulus detection thresholds (painful and non-painful, whether mechanical, vibratory or thermal) and/or dynamic parameters such as temporal summation or conditioned pain modulation [5,6,7]. It is necessary to emphasize that an altered result in these tests is not pathognomonic of NP; however, they may be of interest when a neuropathy has not been identified with a conduction study (which assesses Aβ fibers) and involvement of Aδ and C fibers is suspected [8].

Even so, these tests have two major limitations. First, the high cost of most of the measuring instruments used makes their use in routine clinical practice difficult, and second, the great variability in mean reference values depending on race, sex and other parameters.

For this reason, affordable and cheap means are currently being investigated to bring quantitative sensory tests closer to routine clinical practice, with some of them showing good validity with respect to reference tests [6].

The above tests are the main screening methods in physiotherapy, but in order to establish a definitive diagnosis of NP, diagnostic tests are needed that truly identify nerve dysfunction (although this does not always involve pain). Imaging tests are useful to detect an underlying neurological disease, but they do not allow us to evaluate the integrity of the somatosensory nerve fibers as such. For this, we have another series of tests. The most used clinically are nerve conduction tests and somatosensory evoked potentials, which allow us to evaluate the integrity of non-nociceptive afferent fibers (Aβ), which can be interesting for diagnosing neuropathy, but these fibers are not always involved in the NP [7,8]. When they are not, nociceptive fibers (C and Aδ) are evaluated using laser-evoked potentials and contact heat-evoked potentials [7,8]. It is also possible to directly assess the intraepidermal nerve fibers through a skin biopsy in case of suspicion of small fiber, metabolic and inflammatory neuropathy [8].

### 1.4. Epidemiology of Neuropathic Pain

The epidemiological study of NP has advanced significantly thanks to the existence of detection questionnaires such as the LANSS or PainDETECT. It is estimated that its prevalence is around 6–10% in the general population depending on the country (around 22% of chronic pain) and that it is more frequent in people over 50–60 years of age, with a predominance of female sex, highlighting the lower limbs and the back as the body areas where it seems to manifest the most [5,9].

Chronic pain is a problem in multiple areas. It is the main cause of work absenteeism and presenteeism with illness (notably chronic low back pain), which in turn leads to a decrease in quality of life and an increase in psychosocial difficulties. NP is no different and its presence in workers is associated with high rates of absenteeism, depression and anxiety, burnout syndrome and generalized pain [9].

### 1.5. Mechanisms Involved in the Experience of Neuropathic Pain

The mechanisms involved in the genesis of NP not only vary depending on the etiology but also depending on the sensory profile. Depending on the clinical subtype of NP, the neurophysiological alterations involved in the pain experience vary. For example, continuous burning pain is related to spontaneous C-fiber activity and peripheral sensitization (although other mechanisms also appear to be involved in some neuropathies). Meanwhile, paroxysmal pain is related to spontaneous activity of Aβ fibers and dynamic mechanical allodynia to peripheral sensitization in addition to, in NP conditions, sensitization in the spinal dorsal horn that favors Aβ fibers that can stimulate nociceptive pathways [7,10].

It is important to highlight the strong relationship between NP and the perpetuation of pain. When a peripheral nerve injury occurs, a series of morphofunctional changes occur in ascending nociceptive pathways, in glial and immune cells (both in the CNS and PNS) and in structures of the descending pain modulation system, among which the rostral ventromedial bulb stands out. The rostral ventromedial medulla (RVM) and the periaqueductal gray matter (PGM) together favor the chronification and facilitation of the pain experience [11,12].

### 1.6. Physical Exercise and Health

The impact of exercise on health is multisystemic. It is known that muscle, beyond being a contractile tissue, is a paracrine and endocrine organ [13]. In this way, muscle signaling secondary to exercise can be key in improving health [14,15]. Focusing on the nervous system, the improvements offered by exercise are also very diverse. Exercise has been shown to be useful in the management of neuropsychiatric diseases and neurodegenerative diseases thanks to its anti-inflammatory, antioxidant, neurogenic and angiogenic effects (among other causes) [15,16,17]. Additionally, recent empirical studies provide modest support for the claims that regular exercise not only enhances physical health but also confers resilience against the negative emotional consequences of stress. In a study involving 111 healthy men and women, those who engaged in regular physical exercise at least once per week exhibited lower resting heart rates compared to non-exercisers, highlighting the cardiovascular benefits of exercise. Furthermore, while habitual exercise did not significantly alter cardiovascular responses to acute psychosocial stressors, it did mitigate the decline in positive effect following stress exposure, suggesting protective effects against stress-induced emotional disturbances [18].

Moreover, the potential effect of exercise on nerve regeneration after peripheral nervous system (PNS) injury is encouraging and supported by preclinical evidence. However, the results in humans, although promising, are not entirely clear [19,20]. Furthermore, exercise offers good results in symptomatic improvement and strength recovery in peripheral neuropathies such as diabetic and chemotherapy-induced neuropathy [20]. In addition to these findings, there is a growing body of research illuminating the complex factors within the central nervous system (CNS) that confer resilience to behavioral stress and pain. Recent studies have begun to identify specific neuronal cell types in the peripheral nervous system and key genes contributing to pain resilience. This improved understanding of resilience factors is pivotal in informing the development of novel treatments for chronic pain and behavioral stress, which might remain undiscovered with a sole focus on disease susceptibility [21].

### 1.7. Physical Exercise and Pain

Nowadays, the role that physical exercise plays in reducing pain is undoubtedly recognized. Regular exercise causes functional changes in structures of the descending modulatory system, producing in turn an activation of the endogenous opioid, serotonergic and cannabinoid systems both in healthy subjects and in pain conditions (including NP), which together induce hypoalgesia [22,23]. However, its effect differs in chronic pain conditions that present with altered conditioned pain modulation and may even have acute hyperalgesic effects when exercise is strenuous [22,23,24]. In turn, regular exercise has a great impact on the functioning of the immune system, inducing an anti-inflammatory state that reinforces exercise-induced hypoalgesia (EIH). These effects occur both at the local site of injury and at the systemic level, and even in the CNS [23]. Although these seem to be the main mechanisms, there are numerous other processes that seem to be associated with EIH, such as the release of myokines and the influence it exerts on the nitric system, thus being an effective therapy in pain conditions of diverse etiology [23].

Furthermore, recent studies highlight the psychological aspects of pain, particularly in acute low back pain (LBP). Pain-related fear and pain catastrophizing have been associated with disability and actual performance in chronic pain patients. In the context of acute LBP, an experimental, cross-sectional study involving 96 individuals demonstrated that pain-related fear, as measured by the Tampa Scale for Kinesiophobia, was the strongest predictor of actual performance in a dynamic lifting task. Additionally, both pain-related fear and pain catastrophizing, as quantified by the Pain Catastrophizing Scale, were found to be significantly predictive of perceived disability, more so than pain intensity itself. These findings underscore the importance of psychological factors like pain-related fear in influencing daily activities of individuals experiencing acute LBP. The clinical implications of these results are significant, especially in developing preventive strategies for chronic LBP [25,26].

### 1.8. Physical Exercise and Neuropathic Pain

The possible influence that exercise has on nerve repair, added to the various mechanisms involved in EIH, justifies the possibility that exercise may have an influence on the improvement of different NP phenotypes.

Research aimed at assessing the influence of exercise on NP has begun to grow since 2011. To date, the most studied thematic categories that relate them are neuroscience and clinical neurology, followed (with considerably fewer publications) by rehabilitation, endocrinology and sports sciences [27]. Research in animal models has a significant number of publications, while research in humans is still deficient. This is why we are currently beginning to see more and more clinical trials investigating hypoxic-ischemic encephalopathy in various conditions of peripheral NP, such as diabetic neuropathy, chemotherapy-induced neuropathy, radiculopathy or sciatica.

The typology of physical activity is determining in terms of the number of published studies, and although most of the studies that evaluate NP in animal models are carried out with aerobic exercise (AE), its therapeutic effectiveness has hardly been evaluated in humans, with the exception of diabetic neuropathy. Although there is no official definition of AE, we could include all those activities whose execution requires oxygen to produce the energy necessary to execute them. The definition requires a review, but in the literature, cycling, swimming, walking and running are mainly accepted within their modalities, including their variants (as long as they are performed at certain intensities).

### 1.9. Justification

Due to the lack of information and context of the influence of AE, in this narrative review we will try to gather scientific evidence that addresses the relationship between AE and NP in peripheral neuropathies of mechanical origin, that is, that which arises from direct mechanical influences such as compression, overstretching or sectioning of the affected peripheral nerve, either partially or totally. Therefore, the topic will be approached assessing the evidence and responsible mechanisms in animal models and in humans.

## 2. Methodology

The search for scientific articles was carried out in the PubMed and SCOPUS databases.

Due to the lack of key terms specific to AE and neuropathic pain, other MeSH or DECS terms such as “neuralgia”, “sciatica”, “sciatic neuropathy”, “nerve compression syndromes”, “exercise”, “exercise therapy”, “endurance training”, “swimming” and “radiculopathy” were used. In addition, the free terms “aerobic exercise”, “neuropathic pain” and “treadmill” were searched. The Boolean operators “AND” and “OR” were used to mix the aforementioned terms. In addition, “NOT” was used with the terms “diabetic”, “diabetes”, “chemotherapy”, “HIV” and “spinal cord injury” to eliminate those articles carried out in the most-studied populations and that did not meet the selection criteria of this review.

Different exclusion criteria were established. Publications prior to 2011 and published in languages other than Spanish and English were discarded. Clinical trials were discarded where AE was not the main variable or were carried out in a population of any kind where NP was not the result of a mechanical force exerted on the nerve, whether compression, section or overstretching. Clinical trials with n = 1 were not included. In animal models, only clinical trials that also included the study of some neurophysiological mechanism related to NP that could be altered by the practice of AE were included. After the search, a total of 19 articles (present in both databases) met the selection criteria, all of them in animal models. In human models, 2 articles were added that included the target population, but not exclusively, due to the lack of articles that relate AE to this population only. In addition, the reference to 7 narrative reviews and 1 book was included to put the results obtained in the neurophysiological measures in context due to the complexity of their understanding. This made a total of 28 articles and 1 book included in the development section (Figure 1).

## 3. Evidence in Animal Models and Responsible Mechanisms

Animal models are commonly used to investigate the mechanisms of EIH under different conditions, NP among them. To study peripheral NP of mechanical origin, different methods of inducing nerve damage are applied. Among them, the most used are partial nerve ligation (a strongly tight constrictive ligation), chronic constriction (several loose ligatures around the nerve), nerve transection (a total transection of the nerve with posterior suturing of the ends) and the neuroma model (a nerve transection without posterior suture to prevent its regeneration) [12]. Even so, there are other, less common methods, such as acute crushing of the peripheral nerve [28]. The presentation and time course of symptoms vary between the models used, as well as the degree and speed of nerve regeneration. For example, in neuroma or nerve transection models it is common to find autotomy behaviors, while these are not common in partial section or chronic constriction models. Similarly, in chronic constriction models it is very common to find hypersensitivity to cold, while after partial ligation there is usually hyporeactivity [12]. Most studies are carried out in rats, and to a lesser extent, in mice. Such studies usually intervene through an injury to the sciatic nerve; however, others can be used, such as the trigeminal nerve [29].

NP is mainly evaluated in two variables:Mechanical allodynia, mainly quantified by Von Frey filaments, a punctate stimulus that is applied to the skin in the territory innervated by a nerve, whether it is the one itself injured (common in models of partial ligation [30] and chronic constriction [31]) or others that are not injured (common in nerve transection models in the initial stages due to the denervation that occurs in the injured nerve), generally in the saphenous nerve in sciatic injury models [32,33,34]. The punctate stimulus is applied through increasing intensity [29] or through temporal summation at the same intensity [35]. Another method used is a pressure algometer of increasing intensity [36].Allodynia or thermal hypersensitivity, generally quantified by measuring the latency time after the emission of radiant heat at a controlled intensity, usually for a maximum of 20 s to avoid damage [36,37]. Some studies also measure hypersensitivity to cold, using cold surfaces (measuring the withdrawal latency time) [28] or acetone injections [36].

In addition to directly measuring NP, many studies perform complementary tests to correlate and, when possible, demonstrate causality of neuroimmunological mechanisms underlying NP and that are involved in EIH. Among them we can find serum measurements in addition to immunohistological measurements in key places of peripheral NP modulation, either in the injured nerve itself (in the dorsal root ganglia (DRG) or in more distal areas), in the spinal cord and in central structures (such as the RVM, PGM, locus coeruleus (LC) and raphe nuclei) [29,32,38].

### 3.1. Potential of Aerobic Exercise in Relation to Mechanical Allodynia

The ability of AE to reduce mechanical allodynia in animal models with peripheral NP of mechanical origin has been widely studied. Of all the clinical trials included, only the study by Rostami et al. [29] showed an exercise group that did not obtain favorable results after the injury in relation to the control group with induced nerve injury (INI). The reason may lie in the fact that the model they carried out was in the trigeminal nerve and not in the sciatic nerve like the majority, or in the sex differences between groups. Those that did not improve were male rats (after an exercise session) while the females did and they improved more than the males after 2 weeks of AE. However, some exclusive studies of male rats obtained favorable results in the group that performed AE [35,37]. Therefore, the influence of sex still needs to be further studied.

Some studies have also assessed which modality of AE is most effective in reducing mechanical allodynia, although most evaluate and highlight the effectiveness of a single form of AE; either no-incline treadmill [28,30,31,32,33,34,37,38,39,40], 8% incline treadmill [41,42,43] or swimming [29,35,44]. In 2012 [45], Chen et al. discussed whether a swimming program and a treadmill program of increasing intensity reduced MA equally in rats, showing hypoalgesia for a longer duration in the swimming group. However, the swimming group did 4 more weekly sessions, so the amount of exercise could have influenced the differences. Tsai K. et al. [46] found that using a treadmill with an 8% incline offered better results than without an incline (performing the same exercise protocol), and in the same way the dose of effort may be the main determinant of the differences (and not so much the modality).

Most studies carry out post-surgical exercise protocols, starting some in the first 7 days after the intervention [28,29,33,34,35,36,38,39,40,41,42,43,44,45,46], obtaining favorable results.

Several studies have assessed whether results are similar using pre-surgical exercise protocols and whether adding them to a post-surgical protocol offers greater improvement. Two of the trials [30,37] carried out a protocol of 2 weeks of pre-surgical treadmill exercise plus 6 days of post-surgical exercise, demonstrating that it could reduce mechanical allodynia (MA) compared to INI from the 3rd–4th day. However, these results were not compared with a postsurgical exercise protocol.

Grace et al. [31] compared the effectiveness of a pre-surgical AE protocol with respect to two post-surgical exercise protocols (one started immediately after the operation and another two weeks later), showing that although all were effective in attenuating mechanical allodynia, in the pre-exercise group the intervention was attenuated earlier than the other two protocols. In line with these results, in 2011 [28] Bobinski et al. demonstrated that a pre-surgical exercise protocol could be effective in reducing MA, although not as effective as a postsurgical one or the sum of both. Overall, these trials suggest that presurgical AE may be effective in reducing mechanical allodynia induced by nerve injury, and it may be interesting to add it to a postoperative AE program to maximize results.

Only one of the included studies has assessed the influence of AE intensity on EIH [43]. The results showed that “high intensity” AE (16 m/min) offered better results than “low intensity” (10 m/min) in attenuating mechanical allodynia. However, they only had 6 members, which limits the veracity of the results.

Putting this in context with the rest of the articles, we can see how the intensity of the protocols (on a treadmill) varied from 7 m/min [30,31,32,33,34,35,36,37], 8 m/min [39] and 10 m/min [28,38,40] to programs that reached 30 m/min [45], all of which have been shown to be effective in reducing mechanical allodynia. Even in trials where animals ran voluntarily [31], AE offered improvements; therefore, with current data it cannot be known which intensity is better. It can be stated, however, that AE at very different intensities continues to be effective.

A few clinical trials have also assessed how EIH varies when other treatments are added to AE. Two trials [41,42] compared EIH mediated by an 8% inclined treadmill exercise protocol with an ultrasound protocol and with the sum of both, demonstrating that although the AE protocol attenuated mechanical allodynia, the effects were better if therapeutic ultrasound was added.

Similarly, Cobianchi et al. [33] compared the effect on mechanical allodynia of a treadmill AE protocol with an electrostimulation program and with the sum of both (after a sciatic nerve transectomy). The results showed how the sum of both therapies results in greater attenuations of mechanical allodynia (in the cutaneous territory innervated by both the saphenous and sciatic nerves), and in an earlier recovery of sensitivity in the cutaneous territory innervated by the sciatic nerve. 

The three studies allow us to conclude that the ideal is to add the AE protocol to other therapies that have scientific evidence to reduce NP in order to obtain better results.

### 3.2. Potential of Aerobic Exercise in Relation to Thermal Hypersensitivity

Heat hypersensitivity was monitored by 12 of the included trials. The results of AE on this variable are not so clear, although the general trend indicates that AE attenuates the thermal allodynia induced by nerve injury [30,31,32,33,34,36,41,42,43,44,45,46].

In nerve transection models, AE does not manage to reduce heat hypersensitivity in the sciatic territory, nor does it recover sensitivity sooner (as happened with tactile stimuli), even if it was added to an electrostimulation program [33]. AE seems to improve heat allodynia in the saphenous territory [33,34], although the group that performed 10 exercise sessions instead of 5 in the 2015 trial by López-Álvarez et al. [34] did not reduce it. However, for some unknown reason this group only took measurements on days 8 and 15 of the entire experiment (which consisted of 60 days), which invalidates any comparison between both exercise protocols. The addition of AE to other treatments such as therapeutic ultrasound or electrostimulation also seems to reduce heat hypersensitivity to a greater extent [33,41,42].

It should be noted that AE-induced changes in blood flow could indeed complicate the interpretation of results when assessing thermal hypersensitivity in the skin of experimental animals. The increase in blood flow to the skin during and after AE can alter the skin’s temperature and its sensory responses, potentially shifting thermal pain thresholds and affecting the measurement of thermal hypersensitivity. Furthermore, AE is known to induce hypoalgesia, a decreased sensitivity to pain, due to the activation of endogenous pain inhibitory systems, which could lead to an underestimation of pain responses to thermal stimuli. Additionally, the anti-inflammatory effects of regular AE might modulate inflammatory responses, influencing pain sensitivity, including thermal hypersensitivity. Thus, these exercise-induced physiological changes should be carefully considered when designing experiments and interpreting results in such studies.

Hypersensitivity to cold was only measured in two of the trials, both nervous constriction models [28,36], in accordance with its high prevalence in these nervous induction models [12]. The results suggest that post-surgical exercise is effective in reversing it [28,36], while pre-surgical AE does not seem to offer this improvement [28].

### 3.3. Immunological and Neuroplastic Changes in the Peripheral Nervous System Involved in Exercise-Induced Hypoalgesia

Various clinical trials dedicate their study to assessing the influence of exercise on different markers of NP at the peripheral nerve level. These studies evaluate it in the DRG [31,33,34,36,39,44], the proximal axonal thickness [28,31,36,40,41,45,46], the distal [31] and in the nerve terminals [34].

#### 3.3.1. Role of Proinflammatory Cytokines and Chemokines

Cytokines are inflammatory mediators that also intervene in the induction, perpetuation and attenuation of peripheral NP. Among them we can find some pro-inflammatories (interleukin 6 (IL-6), interleukin 1 beta (IL-1β), tumor necrosis factor alpha (TNF-α), etc.) that increase after nerve damage [44], and other anti-inflammatories (such as interleukin 4 (IL-4) and interleukin 10 (IL-10)) that reduce the acute period after injury [42,46]. At the level of the nervous lesion focus and in the DRG, proinflammatory cytokines are expressed by glial cells (Schwann cells), which in turn promote greater release by macrophages, neutrophils, mast cells and T lymphocytes (among others) [47], ultimately generating afferent sensitization and an increase in ectopic discharges along the nerve, especially in the DRG, which contribute to central sensitization of the spinal dorsal horn [12].

IL-1β increases acutely, favoring local demyelination, the recruitment of immune cells (especially macrophages) and the release of sensitizing substances (substance P, prostaglandins, etc.), excitatory neurotransmitters and neurotrophic factors by sensory neurons [12,47]. It is even capable of directly stimulating nociceptive afferent neurons, like TNFα [11,12]. AE reduces the increase in IL-1β after nerve damage, both after an exercise session [45] and especially with regular AE [28,42]; furthermore, it is capable of increasing IL-1ra (a cytokine that prevents the binding of IL-1β to its receptor) [40].

On the other hand, injury produces an increase in plasma IL-1β so that it reaches the CNS and activates glial cells [47]. Although the influence of AE on blood levels of IL-1β has not been measured, one ex vivo experiment in monoclonal blood cells (monocytes and lymphocytes) showed that exercise is capable of attenuating the increase in IL-1β in them after the injury [31].

IL-6 and TNFα induce sensitization of nociceptive pathways, favor the recruitment of immune cells and are directly related to NP [12,47]. Regarding TNFα, post-surgical AE is capable of reducing the levels of this cytokine in the injured nerve [28,41,45,46], which does not occur with pre-surgical AE [28]. The increase in IL-6 in the PNS is attenuated by 8% incline treadmill exercise [42,46], but not without incline [40,46], although in the trial by Bobinski et al. [40] this was to be expected, since for some unknown reason IL-6 levels did not increase in the INI. The same authors evaluated the influence of AE (both pre-surgical and post-surgical) on the levels of IL-6 receptors in the nerve [28] showing that AE attenuated the increase in the expression of these receptors induced due to nerve damage.

Chemokines are also involved in the immune response and in the generation of NP following peripheral nerve injury. After damage, their release by Schwann cells increases to generate a powerful chemoattraction of immune cells, including macrophages, which are also responsible for releasing them [12,47,48]. Grace et al. [31] showed how AE very significantly attenuated the increase in the C-C motif chemokine 2 ligand (CCL2) both in the DRG and in the axonal fibers proximal and distal to the lesion. Furthermore, they monitored the influence of AE on serum chemokines (CCL2, CCL3 and the C-X-C motif chemokine 1 ligand (CXCL1)), showing that although INI greatly increased their levels, AE managed to maintain their levels similar to those of the non-operated control groups.

Stem cell Wnt signaling pathways are involved in the pathogenesis of NP. Wnt3a interacts with intracellular receptors (such as low-density lipoprotein receptor-related proteins 5 (LRP5) and 6 (LRP6)), triggering a cascade reaction that leads to the migration of the β-catenin protein to the cell nucleus, favoring the transcription of algogenic substances (such as cytokines and brain-derived neurotrophic factor, BDNF) and synaptic plasticity. Furthermore, it is related to the central sensitization characteristic of NP by inducing microglial activation in the dorsal horn [49].

In 2021, Cho et al. [39] studied the influence of exercise on the Wnt/βcatenin signaling pathway in DRG neurons after chronic crushing of the sciatic nerve, showing how AE attenuated the activity of this pathway through a reduction in levels of Wnt3 and those of its receptor LRP6, events that resulted in a lower translocation of β-catenin to the cell nucleus. Thus, although it is too early to say, EIH may be mediated by the inhibition of this signaling pathway.

#### 3.3.2. Role of Macrophages and Anti-Inflammatory Cytokines

Macrophages are immune cells that participate in the neuroimmune response after peripheral nerve injury. These are activated, in part, by the damaged neurons themselves through various released compounds (substance P, adenosine triphosphate (ATP), CCL2, etc.). Some come from plasma monocytes, while others come from populations resident in the SNP itself [48].

These can be subdivided based on the functional phenotype they present into M1 macrophages (pro-inflammatory in nature) and M2 macrophages (anti-inflammatory in nature), the M1/M2 ratio being upregulated after nerve injury. M1 macrophages release substances such as cytokines (TNFα, IL-1β, IL-6), neurotrophic factors (NGFs) and chemokines (CCL2, CXCL1), thus sensitizing the primary afferent fibers. In contrast, M2 macrophages attenuate this hyperexcitability by releasing anti-inflammatory substances such as IL-4, IL-10 and IL-13 [48].

The influence of AE on macrophages has been assessed by two of the included trials, with somewhat disparate results. In 2016, Grace et al. [31] showed that AE reduced the number of macrophages compared to INI, both total and M1 and M2 evaluated separately. Interestingly, AE reduced the number of total macrophages proximal to the lesion, but not distal to it. These results, added to the decrease in cytokines (anti-inflammatory and pro-inflammatory) shown in their experiment in monoclonal blood cells after AE, suggest a hyporeactivity of macrophages in general, which could justify the EIH in their study despite the reduction in M2 macrophages. Even so, these results differ from others where AE attenuated the decrease in the levels of anti-inflammatory cytokines IL-4 [40], IL-1ra [40] and IL-10 [41,46] after injury (in the PNS), which suggests a maintenance of M2 macrophages.

Bobinski et al. [40] assessed the influence of AE on the phenotypic proportion of macrophages in the injured nerve. INI increased the ratio of M1/M2 macrophages very notably; however, exercise maintained this proportion similar to that of non-operated mice, suggesting that AE favors the anti-inflammatory action of macrophages. The authors themselves highlight that the results of both studies may differ because they are measuring proportions of macrophages, and not quantity, as the previous one did. Furthermore, in this study the AE program was post-surgical, while in the previous trial it was pre-surgical, which may be decisive.

In this same trial, Bobinski et al. [40] more closely evaluated the influence of exercise on IL-4 levels and its relationship with mechanical allodynia and the phenotypic expression of macrophages. They managed to show how EIH was dependent on the expression of IL-4. Furthermore, they showed that IL-4 mediates the phenotypic change from M1 to M2 macrophages caused by AE.

#### 3.3.3. Role of Neurotrophins

Neurotrophins, notably BDNF and nerve growth factor (NGF), are neurotrophic factors involved in neuronal survival and plasticity processes, with diverse roles in both the PNS and CNS. These neurotrophins, secreted by neurons, immune cells and Schwann cells (mainly in the DRG), bind to receptors like p75 and the tyrosine kinase family receptors TrkA (for NGF), TrkB (for BDNF) and TrkC. Through these interactions, they modulate various signaling pathways, including those that promote nociceptive sensitization [12,50].

NGF, known to increase after nerve injury, induces phenotypic changes in synaptic terminals and escalates the release of pronociceptive mediators like BDNF and substance P, contributing to the sensitization of nociceptive pathways and ectopic hyperactivity of primary afferent fibers. NGF also plays a role in collateral nerve fiber formation post injury, potentially leading to allodynia expansion [12,50].

BDNF, similarly upregulated after nerve injury, is promoted by NGF production and mechanisms like the activation of the Wnt/β-catenin pathway. It is then anterogradely transported to dorsal horn postsynaptic terminals, where, along with BDNF from other cells, it sensitizes the nociceptive pathway [12,49,50]. However, it is essential to note that the role of BDNF, particularly in relation to AE, is complex and multifaceted. While increased peripheral BDNF is implicated in sensitizing the pain pathway, substantial literature indicates that AE elevates plasma BDNF levels in healthy individuals and chronic pain patients, a phenomenon associated with beneficial effects like enhanced cardiovascular health and mood improvement [51,52]. This suggests that the influence of BDNF on pain pathways may differ based on context and physiological state.

Recent RNA sequencing data from the human spinal cord further challenges the understanding of BDNF’s role in pain, showing no BDNF expression in microglia, thus contradicting the notion of BDNF release from human microglia in pain modulation [53]. This finding underscores the need for a nuanced approach in interpreting BDNF’s role, considering its diverse functions and context-dependent effects.

The influence of AE on neurotrophin expression post-injury has been assessed by studies showing that activities like swimming and treadmill exercise attenuate BDNF and NGF increases at injury sites and DRGs, suggesting a potential moderating effect of AE on neurotrophin-mediated nociceptive processes [34,35,44]. In 2013, Cobianchi et al. [33] measured the influence of AE with or without electrostimulation on BDNF and NGF mRNA, finding that AE can inhibit the gene expression of these neurotrophins in the DRG post injury, highlighting the complex interplay between AE, neurotrophin expression and pain modulation.

### 3.4. Immunological and Neuroplastic Changes in the Spinal Cord Involved in Exercise-Induced Hypoalgesia

The medullary dorsal horn is a key site in nociceptive processing. Different types of neurons and glial cells interact to ultimately facilitate or inhibit ascending nociceptive transmission. In NP models, various functional changes occur in these cell groups, the results of which converge in central sensitization at this level [12].

#### Role of Central Glial Cells

Classically, it was thought that these cells only provided support in the CNS, but now it is known that they can interact with each other and with neurons, intervening in processes such as the maintenance of homeostasis, neural survival and nerve conduction in the CNS. Among them, we are going to focus on microglia and astrocytes due to their strong relationship with NP.

Microglia are a resident population of macrophages in the CNS. They are activated after nerve injury, changing from amoeboid to hypertrophic conformation [12,34]. Then, they accumulate around the neurons of the dorsal horn to release various substances that will contribute to the pathogenesis of NP [48], among them BDNF, IL-1β and TNFα [12]. The typical persistence of NP is mediated (in part) by chronic microgliosis resulting from constant ectopic activity of primary afferent fibers [12].

The influence of AE on microglial levels and activity has been assessed in several trials. AE (treadmill [34,40,42] and swimming [44]) attenuates the increase in spinal microglia secondary to nerve injury, even more so when combined with an ultrasound program [42]. Furthermore, AE not only reduces microgliosis but also prevents the conformational change of these cells (from amoeboid to hypertrophic) [34], which translates into less microglial activation.

In contrast to these results, in 2016 Kami et al. [37] did not find a reduction in total spinal microglia in the exercised group compared to the INI after nerve injury. Even so, it was able to reduce the levels of histone deacetylase 1 (HDAC1)-positive microglia (involved in the expression of algogenic substances released by microglia).

BDNF released in the dorsal horn after injury, although partly derived from DRGs, mainly comes from spinal microglia [12,50]. Its sensitizing action on nociceptive secondary afferent neurons occurs through different mechanisms.

On the one hand, it interacts with TrkB by deregulating the cotransporters Na-K-Cl 1 (NKCC1) (which increases the concentration of intracellular chloride) and K-Cl 2 (KCC2) (which decreases the concentration of intracellular Cl-) in nociceptive neurons, increasing and decreasing them, respectively. This conditions the function of GABAergic inhibitory neurons (dependent on the previous Cl- concentration), meaning that they can act by facilitating these neurons [12,50]. Furthermore, after the injury there is a loss of GABAergic neurons and a decrease in the levels of gamma-aminobutyric acid (GABA) and enzymes involved in its synthesis from the glutamate decarboxylase family (GAD65 and GAD67), which together results in a loss of inhibition [12,30,36].

On the other hand, BDNF increases the phosphorylation of N-methyl-D-aspartate acid receptors (NMDAr) [50], which is added to a decrease in glutamate transporters 1 (GLT-1) (involved in its reuptake into the intracellular space, mainly glial) [31]; both findings facilitate glutamatergic transmission in nociceptive pathways.

Astrocytes are also involved in peripheral NP. These maintain a decisive communication with the microglia, releasing cytokines that activate them and, in the same way, the microglia releasing cytokines that activate the astrocytes. They are capable of inactivating KCC2 channels to a certain extent by increasing extracellular glutamate, facilitating the function of NMDAr, releasing chemokines (CCL2, CXCL1, etc.) and promoting the synthesis of BDNF by microglial cells. This last mechanism does so (in addition to the increase in proinflammatory cytokines) through an increase in the release of ATP to the extracellular space, which binds to the P2X purinoreceptors of spinal microglia (which are also upregulated), activating them [54].

The influence of AE on the number of total astrocytes has been less studied. Only two of the included trials measured it, showing how AE reduces the total number of astrocytes after injury [40,44], although not those specific ones positive in HDAC1 [37].

Other indirect markers of microgliosis and astrocytosis have been measured in these trials to evaluate the influence of exercise on them. The increase in proinflammatory cytokines seems to be attenuated by the practice of AE (IL-1β [28], TNFα [28] and IL-6 [42]), as well as that of the IL-6 receptor [28], although in the trial by Bobinski et al. 2018 [40], AE did not reduce IL-6 levels in the dorsal horn. Similarly, AE seems to reverse the decrease in anti-inflammatory cytokines in the spinal cord that causes nerve injury (IL-4 [40], IL-1 [40] and IL-10 [28]).

Several of the changes that occur in the function of GABAergic inhibition are prevented by AE, such as the loss of GABAergic neurons [30] and the levels of GABA [30] and its glutamate decarboxylase family transporters GAD65 [30,36] and GAD67 [30]. Furthermore, AE seems to be able to reverse the loss of the inhibitory function of these interneurons by attenuating the decrease in KCC2 channels in the dorsal horn and the increase in NKCC1 channels in the DRG [34].

It should be noted that AE can also attenuate microglial activation induced by astrocytes by dampening the increase in expression of P2X family channels in the medulla [31,36] and reduce glutamatergic transmission by attenuating the decrease in GLT-1 in the dorsal horn [31]. Furthermore, it has been shown to attenuate the increase in transcription factors involved in the synthesis of sensitizing substances and their receptors at the spinal level [31,44] (Figure 2).

### 3.5. Changes in Descending Pain Modulating Systems in Neuropathic Pain Models and the Influence of Exercise

Monoaminergic systems are involved in the central modulation of pain. Monoamines are released as neurotransmitters from nuclei in the brain stem. Among them, the most studied in NP models are serotonin and norepinephrine. Each of these systems is expressed by axonal projections from specific central nuclei, which vary depending on the monoamine. For example, the LC stands out as a pain modulator due to its noradrenergic projections, while the RVM and the raphe nuclei stand out for their serotonergic projections. The function of these monoamines as facilitators or inhibitors of pain depends on the receptors and neurons on which they interact. For example, if they stimulate an inhibitory neuron or inhibit a nociceptive neuron, their function will be analgesic, while if they stimulate a nociceptive neuron or inhibit an inhibitory interneuron, their function will be hyperalgesic. In NP conditions, a series of changes occur that favor the perpetuation of pain. For example, “ON” cells are facilitated and “OFF” cells are inhibited in the RVM, which excites nociceptive pathways. Likewise, the expression of the different receptors causes changes both in the dorsal horn and in the brainstem structures, resulting in an alteration of the descending modulatory system in pain control [11,12,55].

It is still not very clear in what specific way AE modulates the monoaminergic system due to the complexity of the different neuronal pathways involved in it, but these changes do seem to be related to EIH in models of NP [32,38].

Regarding the serotonergic system, AE seems to enhance it in a general way after nerve injury. In NP models, it is capable of increasing serotonergic receptors in brain stem structures (2A [32,38] and others such as 2C, 3 and 7 [38]) and in the dorsal horn (2A [32]); it has even been proven that EIH is dependent on the action of 5HTr 2A [32] and serotonin synthesis in general [38]. Furthermore, AE increases serotonin levels and reduces those of transporters that reuptake it in the brainstem [38].

The involvement of the noradrenergic system in the EIH of peripheral NP models seems to play a more secondary role. AE increases the number of adrenergic receptors (α1, β2) in the dorsal horn and in some brainstem structures [32]; however, the implication of these changes in EIH is not so clear. Inhibition of β2 receptors in LC does not appear to reduce EIH directly, although it does attenuate the reduction of microgliolysis induced by AE at this level, which may secondarily influence NP [38]. General inhibition of β2 receptors was shown to completely inhibit EIH in the López Álvarez et al. trial [32], while general inhibition of norepinephrine synthesis did not reduce EIH in the trial by Bobinski et al. [38]. Therefore, no clear conclusions can be drawn.

The influence of AE on other descending pain modulating systems has also been studied, such as the opioid system, a strong inhibitor of nociception. Endorphins function as neurotransmitters in this system and are increased by AE following nerve injury to key structures of the brain stem. Furthermore, EIH depends largely on the action of this system in the CNS, especially in higher structures [43].

## 4. Evidence in Human Models

The study of the influence of AE in models of peripheral NP of mechanical origin is nonexistent. Several clinical trials in this population have included AE, but not as a primary variable. For example, the 2015 trial by Moustafa et al. [56] was conducted in patients with lumbosacral radiculopathy, and they compared a “functional restoration program” (which included AE in the last weeks) with this same program added to corrective postural exercises. However, no conclusions can be drawn about the effectiveness of this exercise modality because both groups performed it and, in addition, numerous other therapies were added in both groups. In the study by Toth et al. [57], a 15-to-60-min AE and stretching program was compared with a therapeutic education program in an adult population with peripheral NP. The results showed a greater reduction in pain in the AE group, suggesting that this may be effective in reducing peripheral NP in humans. However, the population included was not specific to neuropathies of mechanical origin, since all types of peripheral neuropathies and polyneuropathies (with very different etiologies) were included and, therefore, the effects of AE cannot be directly extrapolated.

## 5. Conclusions

AE has been shown to be effective in reducing NP in animal models with mechanically induced neuropathies. It has also been proven that these improvements result from the attenuation of various neurophysiological mechanisms that are altered in NP models and that influence nociceptive hyperexcitability and pain processing (summarized in Figure 1). However, these results cannot be directly extrapolated to humans for various reasons. One of them is the need for longer recovery times in humans due to the difference in the length of peripheral nerves. On the other hand, the results in rats and/or mice do not always coincide with those obtained in humans. Furthermore, these AE protocols begin in animal models, as already mentioned, within the first 7 days after the intervention, a very short latency time that does not occur in humans after nerve surgery. Finally, the majority of peripheral neuropathic disorders that are treated in physiotherapy have an evolution of progressive damage, not acute (as is the case of the nerve lesions induced in these animal models), which can make the mechanisms involved and the EIH not be similar.

The aforementioned human trials do not allow conclusions to be drawn about the therapeutic potential of AE in models of peripheral NP of mechanical origin and, although this exercise modality has been shown to be effective in other models of NP, the results cannot be extrapolated due to the etiological diversity.

The results in animal models are really promising and, furthermore, the mechanisms involved in EIH are multiple. All of this suggests that the results may be similar in humans. Even so, the lack of trials in human models makes it impossible to draw clear conclusions. Therefore, researchers are urged to evaluate the effectiveness of AE in the population reviewed in this work.

## Figures and Tables

**Figure 1 biomedicines-11-03174-f001:**
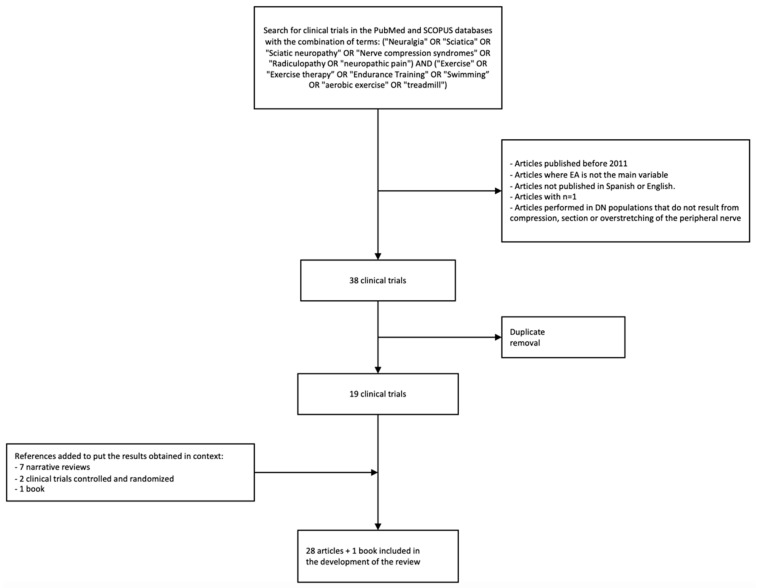
Article selection flowchart.

**Figure 2 biomedicines-11-03174-f002:**
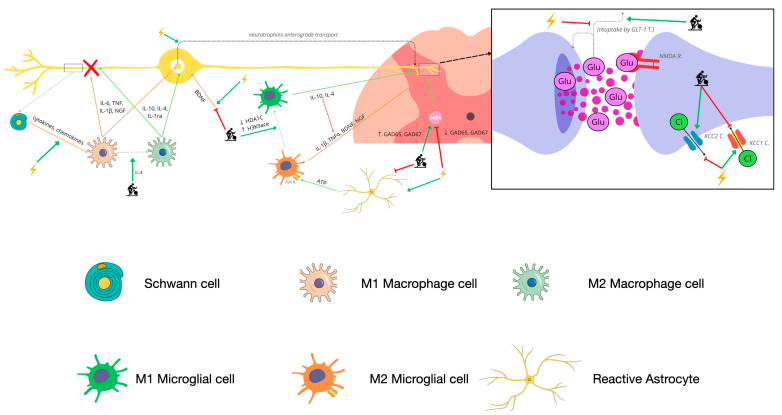
Summary of the mechanisms involved in exercise-induced hypoalgesia at the level of the peripheral nerve and the spinal dorsal horn in models of peripheral neuropathic pain of mechanical origin. Self-made figure with Biorender.

## Data Availability

All data derived from this study are presented in the text.

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
