# Peer review of "Aerobic Exercise and Neuropathic Pain: Insights from Animal Models and Implications for Human Therapy"

_biomedicines, 2023, doi:10.3390/biomedicines11123174_

Round 1

Reviewer 1 Report

Comments and Suggestions for Authors

1.     Did you spell the terms of DN, AD, MA, and CC ?

2.     Please summarize classification of pain and neurofibers pertaining to each pain in the table.

3.     Please the table which states the role and function, the association with neuropathic pain, responsiveness to aerobic exercise in each cytokein, chemokine, macrophage, various factors including BDNF, NGF, NKKCC, and NMDAr, glia, monoamines, and serotonin.

4.     Which tract inhibits ascending nociceptive transmission in the medulla ?

Comments on the Quality of English Language

relatively well/

Author Response

We would like to thank the reviewers for their comments, which we believe have clarified many aspects of the manuscript. We have edited the text according to the suggestions from the reviewers. We have highlighted all changes in yellow throughout the manuscript. A point-by-point response is presented below.

  1. Did you spell the terms of DN, AD, MA, and CC ?

Response: We apologize, but some of the terms were not properly translated to English. We defined the following terms: DN NP (Neuropathic Pain), AD AE (Aerobic Exercise), MA (mechanical allodynia), C-C (type of motif chemokine).

  1. Please summarize classification of pain and neurofibers pertaining to each pain in the table.

Response: We appreciate your suggestion. However, upon careful consideration, we would like to clarify that our article does not currently feature such a table. More importantly, the primary focus of our paper is on exploring the impact of aerobic exercise on neuropathic pain, particularly its neurophysiological mechanisms and direct influence on the pain experience. While we recognize the importance of understanding the various classifications of pain (nociceptive, nociplastic, and neuropathic), our article does not delve deeply into these classifications, as they are not the central theme of our research.

Given this context, we believe that the brief introduction provided in our article sufficiently contextualizes the subject for our readers, orienting them towards the specific nuances of neuropathic pain relevant to our study. This introduction serves to lay the groundwork for the subsequent detailed exploration of the relationship between aerobic exercise and neuropathic pain.

  1. Please the table which states the role and function, the association with neuropathic pain, responsiveness to aerobic exercise in each cytokein, chemokine, macrophage, various factors including BDNF, NGF, NKKCC, and NMDAr, glia, monoamines, and serotonin.

Response: We understand that this is a request for a comprehensive table that details the role and function of various biological factors in the context of neuropathic pain and their responsiveness to aerobic exercise. Our manuscript primarily focuses on the broader neurophysiological effects of aerobic exercise on neuropathic pain. The detailed biochemical interactions of the myriad of factors you mentioned, while undoubtedly valuable, may extend beyond the targeted scope of our research. In addition, the comprehensive analysis of each of these factors in the context of neuropathic pain and their responsiveness to aerobic exercise constitutes a vast and complex area of study. Presenting this information in a single table might overwhelm the primary narrative of our research and potentially detract from its specific findings. Finally, the interactions of these factors in the context of neuropathic pain and exercise are an emerging area of research. As such, there may be limitations in the available data, which could lead to an incomplete or potentially misleading representation in the table.

  1. Which tract inhibits ascending nociceptive transmission in the medulla?

Response: We clarified as follows: “The medullary dorsal horn is a key site in…”

Reviewer 2 Report

Comments and Suggestions for Authors

The authors have written a review on the influence of aerobic exercise on peripheral neuropathic pain with special focus on peripheral neuropathies of mechanical origin. It turned out that very limited literature exists on human clinical studies with regard to pre-defined search criteria. Instead, the authors reviewed animal model data that is hoped to support human relevance.

I have following recommendations and questions:

Large literature supports the idea that aerobic exercise elevates mood, increases stress resilience/coping and reduces anxiety i.e. exercise alleviates typical symptoms/risk factors that are thought to facilitate pain chronification and maintenance of chronic pain (e.g. PMID 24822048, 32976800). Could you discuss this briefly?

Acute low back pain patients are reported to suffer also from kinesiophobia i.e. from fear of movement/exercise (PMID: 16359797). Could kinesiophobia contribute to unwillingness of neuropathic pain patients to exercise? Please discuss briefly.

Would you think that aerobic exercise-induced blood flow changes could complicate interpretation of results obtained assessing thermal hypersensitivity from the skin of an experimental animal?

In lines 486-490 it is discussed that increased peripheral BDNF sensitizes the pain pathway. Large literature in healthy volunteers and chronic pain patients show that aerobic exercise increases plasma BDNF level, a phenomenon that is thought to underline beneficial effects of exercise (PMID: 27658238). BDNF is vital for regulation of heart contraction and relaxation (PMID:25583515). It simply do not make sense that exercise-induced plasma BDNF sensitizes the pain pathway similar to NGF. New RNA seq data from human spinal cord do not show BDNF expression in microglia and do not support the idea that BDNF is released from human microglia (Line 535, PMID: 36731429). Please, try to provide balanced view of the literature.

Comments on the Quality of English Language

Minor editing of english needed

Author Response

We would like to thank the reviewers for their comments, which we believe have clarified many aspects of the manuscript. We have edited the text according to the suggestions from the reviewers. We have highlighted all changes in yellow throughout the manuscript. A point-by-point response is presented below.

The authors have written a review on the influence of aerobic exercise on peripheral neuropathic pain with special focus on peripheral neuropathies of mechanical origin. It turned out that very limited literature exists on human clinical studies with regard to pre-defined search criteria. Instead, the authors reviewed animal model data that is hoped to support human relevance.

Response: Thank you for your feedback.

Large literature supports the idea that aerobic exercise elevates mood, increases stress resilience/coping and reduces anxiety i.e. exercise alleviates typical symptoms/risk factors that are thought to facilitate pain chronification and maintenance of chronic pain (e.g. PMID 24822048, 32976800). Could you discuss this briefly?

Response: Thank you for your recommendations. We added these references and discussed it in the subheading 1.6 (Physical Exercise and Health).

Acute low back pain patients are reported to suffer also from kinesiophobia i.e. from fear of movement/exercise (PMID: 16359797). Could kinesiophobia contribute to unwillingness of neuropathic pain patients to exercise? Please discuss briefly.

Response: Thank you for the cite recommendation. It was added in the subheading 1.7 (Physical exercise and pain).

Would you think that aerobic exercise-induced blood flow changes could complicate interpretation of results obtained assessing thermal hypersensitivity from the skin of an experimental animal?

Response: Aerobic exercise-induced changes in blood flow could indeed complicate the interpretation of results when assessing thermal hypersensitivity in the skin of experimental animals. The increase in blood flow to the skin during and after aerobic exercise can alter the skin's temperature and its sensory responses, potentially shifting thermal pain thresholds and affecting the measurement of thermal hypersensitivity. Furthermore, aerobic exercise is known to induce hypoalgesia, a decreased sensitivity to pain, due to the activation of endogenous pain inhibitory systems, which could lead to an underestimation of pain responses to thermal stimuli. Additionally, the anti-inflammatory effects of regular aerobic exercise might modulate inflammatory responses, influencing pain sensitivity, including thermal hypersensitivity. Thus, these exercise-induced physiological changes should be carefully considered when designing experiments and interpreting results in such studies.

In lines 486-490 it is discussed that increased peripheral BDNF sensitizes the pain pathway. Large literature in healthy volunteers and chronic pain patients show that aerobic exercise increases plasma BDNF level, a phenomenon that is thought to underline beneficial effects of exercise (PMID: 27658238). BDNF is vital for regulation of heart contraction and relaxation (PMID:25583515). It simply do not make sense that exercise-induced plasma BDNF sensitizes the pain pathway similar to NGF. New RNA seq data from human spinal cord do not show BDNF expression in microglia and do not support the idea that BDNF is released from human microglia (Line 535, PMID: 36731429). Please, try to provide balanced view of the literature.

Response: Thank you for pointing out this issue. We have clarified using the references proposed.

Reviewer 3 Report

Comments and Suggestions for Authors

The authors attempted to present in the review article the proof of the effectiveness of the treatment with aerobic exercises in patients with neuropathic pain. They state that the proof for this issue was presented mostly in populations with neuropathic pain secondary to diabetic and chemotherapy-induced neuropathy, contrary to the data presented in their article. The authors aimed to combine the evidence in human and animal models about aerobic exercise  (what? Line 26 in Abstract, to be effective in the treatment?) in peripheral neuropathic pain of mechanical origin, from a perspective of neurophysiological mechanisms and their direct influence on the experience of neuropathic pain. They concluded that in humans (patients under treatment? line 30), the influence of aerobic exercise has not been assessed as the main variable in this particular condition of peripheral neuropathic pain (unclear).

The article probably contributes a lot to present the above issue, the intentions of the authors are significant,  the concept is good, and the number of analyzed and cited references is impressive, but the style of the English language and a large number of narrative abbreviations leave much to be desired. The article should be edited with the help of an English native speaker, the physician would be the best, to be well understood in each of the chapters presented, starting from the Abstract (see some samples above). In its current form, I think that the paper does not meet the criteria to be published in Biomedicines, and needs editorial corrections, although its content is scientifically sound.

Minor revisions

Introduction

Line 38 : Only by physiotherapists? Are the physicians of other specialties not interested in pain treatment?

Line 50 : … nociplastic pain (encompassing conditions not distinctly nociceptive or neuropathic, such as fibromyalgia) [2]…. Not true to the end, please define and clarify better according to IASP.

Line 52: …While this classification is rooted in the foundational biological mechanisms, it does not dictate a specific diagnosis… You need the functional test results to clarify the diagnosis…

Line 120: …except SSEP, which nerve neural conduction tests? Explain, abbreviate, and provide refs. please. Have you heard about the Semmes-Weinstein monofilaments test? You write about von Frey’s filaments in the other parts of the article.

The chapter "Physical Exercise and Neuropathic Pain" was written very briefly, although, as the authors promised in the Abstract section, it was intended to present the main focus of the work.

Text in lines 287-307 needs rewriting and correcting the refs. errors. Similarly, in lines 316-321, some editorial errors need corrections.

References

I think that citation of internet websites is a good source of scientific knowledge. The authors should choose the refs. from the PubMed database mainly with DOI.

The style of citations is in some refs. (e.g. 12) is incomplete, in general, it is far away from the MDPI style.

Comments on the Quality of English Language

English needs extensive revisions

Author Response

We would like to thank the reviewers for their comments, which we believe have clarified many aspects of the manuscript. We have edited the text according to the suggestions from the reviewers. We have highlighted all changes in yellow throughout the manuscript. A point-by-point response is presented below.

The authors attempted to present in the review article the proof of the effectiveness of the treatment with aerobic exercises in patients with neuropathic pain. They state that the proof for this issue was presented mostly in populations with neuropathic pain secondary to diabetic and chemotherapy-induced neuropathy, contrary to the data presented in their article. The authors aimed to combine the evidence in human and animal models about aerobic exercise (what? Line 26 in Abstract, to be effective in the treatment?) in peripheral neuropathic pain of mechanical origin, from a perspective of neurophysiological mechanisms and their direct influence on the experience of neuropathic pain. They concluded that in humans (patients under treatment? line 30), the influence of aerobic exercise has not been assessed as the main variable in this particular condition of peripheral neuropathic pain (unclear).

Response: Thank you for your contributions to improve the clarity and quality of this review. We modified the abstract to provide a better overview of this narrative review. 

The article probably contributes a lot to present the above issue, the intentions of the authors are significant, the concept is good, and the number of analyzed and cited references is impressive, but the style of the English language and a large number of narrative abbreviations leave much to be desired. The article should be edited with the help of an English native speaker, the physician would be the best, to be well understood in each of the chapters presented, starting from the Abstract (see some samples above). In its current form, I think that the paper does not meet the criteria to be published in Biomedicines, and needs editorial corrections, although its content is scientifically sound.

Response: Thank you for your recommendations. We consulted a native speaker and made all the corrections he considered necessary. We hope the current version to solve this English language issue.

Line 38 : Only by physiotherapists? Are the physicians of other specialties not interested in pain treatment?

Response: Certainly no. We did not express correctly our idea and we changed the sentence for better comprehension. Initially, we wanted to declare pain, along with disability, as one of the most important complaints for physiotherapists in our clinical practice. As you pointed out, pain should be multidisciplinary approached due to its complexity.

The new sentence is as follows: “Pain is a predominant challenge for clinicians and researchers across various specialties…”

Line 50: … nociplastic pain (encompassing conditions not distinctly nociceptive or neuropathic, such as fibromyalgia) [2]…. Not true to the end, please define and clarify better according to IASP.

Response: We provided a more adjusted definition of nociplastic pain: “pain originating from changed nociception, even in the absence of apparent or imminent tissue harm that would activate peripheral nociceptors, or without any signs of disease or lesions in the somatosensory system that could cause the pain”.

If the reviewer referred that the example provided is not correct (FMS), we totally disagree. Not only the IASP recognize FMS as a nociplastic pain condition (https://www.iasp-pain.org/centralized-nociplastic-pain-causing-fibromyalgia-an-emperor-with-no-cloths/), also previous research pointed out this issue (e.g., 10.1093/pm/pnac121). However, since this review aimed to focus on neuropathic pain, we deleted the example.

Line 52: …While this classification is rooted in the foundational biological mechanisms, it does not dictate a specific diagnosis… You need the functional test results to clarify the diagnosis…

Response: We corrected the paragraph as suggested.

Line 120: …except SSEP, which nerve neural conduction tests? Explain, abbreviate, and provide refs. please. Have you heard about the Semmes-Weinstein monofilaments test? You write about von Frey’s filaments in the other parts of the article.

Response:      

The chapter "Physical Exercise and Neuropathic Pain" was written very briefly, although, as the authors promised in the Abstract section, it was intended to present the main focus of the work.

Response: The section was intentionally concise to maintain a focused scope on the core aspects of the relationship between physical exercise and neuropathic pain. Our aim was to present the most critical and directly relevant findings without overextending into tangentially related topics. Despite its brevity, the section is built on a comprehensive review of existing literature. We carefully selected studies that provide the most significant insights into the effects of physical exercise on neuropathic pain, ensuring that the content is both relevant and informative. Thus, the content of the "Physical Exercise and Neuropathic Pain" chapter should be considered in the context of the entire manuscript. Key points relevant to this topic are also discussed in other sections, such as the effects of exercise on health and pain modulation. This integrated approach ensures a holistic understanding without unnecessary repetition. We believe that our approach prioritizes clarity and precision in presenting complex scientific information and the succinct nature of the section enhances readability and allows readers to grasp the essential information without being overwhelmed by excessive detail.

Text in lines 287-307 needs rewriting and correcting the refs. errors. Similarly, in lines 316-321, some editorial errors need corrections.

Response: You are correct. We have thoroughly reviewed the entire manuscript to identify and correct all reference and editorial errors.

I think that citation of internet websites is a good source of scientific knowledge. The authors should choose the refs. from the PubMed database mainly with DOI. The style of citations is in some refs. (e.g. 12) is incomplete, in general, it is far away from the MDPI style.

Response: We revised the citation style.

Round 2

Reviewer 2 Report

Comments and Suggestions for Authors

The authors have carefully addressed all points raised by the referees. I would like to thank the authors for writing an interesting and timely review.

Reviewer 3 Report

Comments and Suggestions for Authors

The authors answered most of my questions and applied the suggested corrections. The grammatical quality of the English language has improved, as has the selection of literature and the style of citation. The article may be published in Biomedicines in its current form.